# UnifiedGT: Exploring the Effective Ingredients of Transformers in Large Graphs

## Abstract

In recent years, transformer models have demonstrated great potential for modeling graph-structured data, and many graph transformers (GT) have been proposed and applied to graph representation learning tasks. However, while GTs are effective, existing GTs have mostly been applied to small graphs, and their critical ingredients for success and the connections among these components when processing large graphs are poorly understood. Through a systematic investigation of using GTs on large graphs, we find that (i) explicit graph structure injection through direct neighbor attention masking is significantly more effective than implicitly using graph structure through positional encoding; (ii) combining a direct neighbor-attended GT with a message-passing graph neural network (MP-GNN) boosts accuracy; and (iii) the FFN acts as a semantic mixer and plays an important role, even though some existing GTs neglect the FFN. As part of our systematic investigation, we break down the design space of state-of-the-art GTs and introduce a modular unified GT framework, called UnifiedGT, which is effective at handling both large-scale heterogeneous and homogeneous graph data. UnifiedGT consists of five major components: (i) graph sampling, (ii) structural prior injection, (iii) attention calculation, (iv) composition of local message-passing and long-range attention, and (v) fully-connected layer. UnifiedGT provides different options for each component, which enables practitioners to create new GT methods that significantly improve accuracy over existing methods. Based on comprehensive experiments using UnifiedGT on the Open Academic Graph, we identify our best-performing method, ParDNTrans (GT with a parallel connected MP-GNN and direct neighbor attention masking), which boosts accuracy by **4.5–5.3%** over the state-of-the-art graph transformer.

## 1 Introduction

In recent years, transformer models have been successfully applied to graph representation learning. The ability of transformers to aggregate information across long contexts has shown great potential in alleviating the limitations of conventional message-passing graph neural networks (MP-GNNs), which are based on aggregating local information. Specifically, graph transformers allow us to tackle the expressivity bottlenecks caused by over-smoothing and over-squashing (Song et al., 2023). It allows information to spread across the graph and can be viewed as passing messages in all nodes.

The design of graph transformers (GTs) mainly focuses on how to leverage graph structure information. Existing GTs have used the following three techniques (Min et al., 2022): (1) using positional/structural encoding to make transformer structure aware (Dwivedi & Bresson, 2020; Kreuzer et al., 2021); (2) incorporating structural bias into the attention matrix (Ying et al., 2021); and (3) using message-passing GNN as auxiliary modules (Wu et al., 2021; Rampášek et al., 2022). While GTs are effective on small to medium-sized homogeneous graphs, there is surprisingly little research that has been conducted on heterogeneous graph (Hu et al., 2020; Yao et al., 2020). As a result, the existing GTs' critical ingredients for success and the connections among these ingredients when processing large and heterogeneous graphs are poorly understood. For example, Hu et al. (2020) demonstrates an attention-only heterogeneous GT design that can obtain reasonable accuracy by only attending to direct neighbors, but it is unclear whether integrating more components into it can boost its performance. Yao et al. (2020) performed specific abstract meaning representation (AMR) graph learning on an extended Levi graph by assigning adjacent matrices of subgraphs split from

each edge type to different attention heads, which is a variant of direct neighbor masking. However, it is still unclear how this approach would perform on general tasks. Furthermore, it remains unexplored how existing GTs deal with the following important issues in graph-structured learning:

1. **Information Heterogeneity**. The aforementioned GT techniques have mainly been shown to be effective on graph representation learning tasks on small homogeneous graphs (Ying et al., 2021; Rampášek et al., 2022), while how to deal with the information heterogeneity is yet to be explored on heterogeneous graphs, where different semantic meaning and features need to be considered.
2. **Heavy Long-Tailed Distribution**. Long-tailed distribution in real-world graphs can introduce bias towards the dominant classes during training. Whether the existing GTs can avoid the bias and how they achieve this is under-explored.
3. **Graph Heterophily**. GT models are usually designed under implicit homophily assumptions, which can be problematic for real-world applications like fraudster detection (Pandit et al., 2007; Dou et al., 2020) where heterophilous connections are common. It is important to explore how GTs can deal with graphs with heterophily where nodes or edges could be class inconsistent with their neighbors.

To address the above questions, we conduct a systematic investigation of using GTs on large heterogeneous graphs. We first break down the design space of state-of-the-art GTs and introduce a modular unified GT framework, called UnifiedGT, which is effective at handling both large-scale heterogeneous and homogeneous graph data. UnifiedGT consists of five major components: (i) graph sampling, (ii) structural prior injection, (iii) attention calculation, (iv) incorporation of local message-passing with long-range attention, and (v) fully-connected layer. Second, we summarize the existing design for each component and extend them to solve the three issues listed above. Specifically, to deal with the information heterogeneity issue, we propose a novel attention mechanism, edge-type-based and node-type-based attention and masking method, cross-type direct neighbor masking, which allows the GT to attend to direct neighbors of various node types. To avoid bias towards the dominant classes during training and to deal with graph heterophily, we introduce three designs incorporating local message-passing with long-range attention, i.e., the prefixed, parallel, and prefixed+parallel connection, and propose ParDNTrans, which combines the parallel MP-GNN and GT with attention masking. Third, we conduct comprehensive experiments to explore and analyze the effectiveness of all of the designs of each component and draw some insightful conclusions. For example, we find that (i) explicit graph structure injection through direct neighbor attention masking is significantly more effective than implicitly using graph structure through positional encoding; (ii) combining a direct neighbor-attended GT with a message-passing graph neural network (MP-GNN) boosts accuracy; and (iii) while some GTs neglect the feed-forward network (FFN), the FFN indeed acts as a semantic mixer and plays an important role. Finally, we present our best method, ParDNTrans (a GT with a parallel connected MP-GNN and direct neighbor attention masking), which boosts accuracy by **4.5-5.3%** over the state-of-the-art graph transformer.

In summary, our technical contributions are as follows:

- We design a unified modular graph transformer framework, UnifiedGT, with configurable components for (i) graph sampling, (ii) structural prior injection, (iii) attention calculation, (iv) incorporation of local message-passing with long-range attention, and (v) fully connected layer, with which we are able to express most of the existing GTs.
- We propose two new attention mechanisms for heterogeneous graph transformer, and cross-type direct neighbor masking, which allows the GT to attend to direct neighbors of various node types.
- We propose to combine the prefixed and parallel design of incorporating local message-passing with long-range attention, which can avoid bias introduced by the heavy long-tailed distribution in real-world graphs.
- We empirically show that our best GTs, ParDNTrans, is able to boost accuracy by **4.5–5.3%** over the state-of-the-art graph transformer.

## 2 Graph Transformer Exploration on Large Hetero-Graphs

We provide a preliminary overview of heterogeneous graphs and transformers in Appendix A.1.

## 2.1 CHALLLENGES AND OBSERVATIONS

The design of most existing GTs mainly focuses on graph representation learning tasks on small homogeneous graphs (Dwivedi & Bresson, 2020; Kreuzer et al., 2021; Ying et al., 2021; Wu et al., 2021; Rampášek et al., 2022). However, large-scale heterogeneous graphs differ from homogeneous graphs in several fundamental ways that introduce unique challenges when applying GTs to them, as introduced below. These unique challenges motivated us to perform an empirical study to explore the power of GTs on large heterogeneous graphs. Based on our comprehensive experimental study, we present a few surprising observations.

- **Challenge 1: Information Heterogeneity.** Existing GT has difficulty in learning heterogeneous information, which is popularly observed in many real-world graphs, such as citation networks. It is necessary to inject the graph priors knowledge, including different nodes and edges that are associated with different semantic meanings and features. However, existing GTs inject graph priors knowledge typically through the concatenation or addition of positional encodings, based on which attention scores are calculated via a dot product operation and elements located next to each other will have higher attention to each other. However, in heterogeneous relation types, elements located next to each other could carry very different meanings, which can render positional encodings less intuitive and potentially less effective. For example, in the citation network, for a paper node, though the author node and field node are both direct neighbors to it when concerning a paper-conference classification task, the field node might be able to provide more valuable information than the author node. A better graph structure injection method is needed.

  **Observation: Explicit graph structure injection via attention masking is significantly more effective than implicit graph structure injection via positional/structural encoding.** In a large graph, there are too many nodes that the transformers can attend to. If we let the GT freely attend to all of the available nodes, it will find irrelevant nodes with high similarity to attend to regardless of the injected positional encoding, which is not beneficial for downstream tasks. Interestingly, we find that direct neighbor attention masking can restrict only certain pairs of nodes (i.e., nodes that are direct neighbors) to calculate pairwise dot products, limiting the number of attention scores that are generated, as shown in Figure 1.

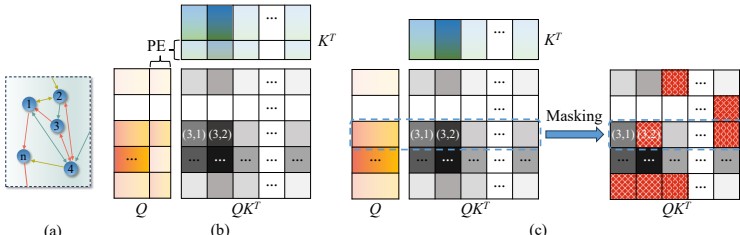

Figure 1: The effect of positional encodings vs. attention masking. A toy example subgraph is shown in (a). The yellow and green matrices $\boldsymbol{Q} \in \mathbb{R}^{n \times d}$, $\boldsymbol{K} \in \mathbb{R}^{n \times d}$ represent the projected node embedding for attention calculation. The gray matrix is the pairwise dot product $\boldsymbol{Q}\boldsymbol{K}^T$, where the gray scale represents the magnitude of the attention. After direct neighbor masking (red cells with a cross pattern) is applied, attention in masked cells are no longer considered. Suppose that we want the information to pass from node 3 to node 1, i.e., the value of element $(3, 1)$ is the attention score we want. (b) The concatenated position encoding (PE) cannot effectively increase the attention at position $(3, 1)$ since node 3 is more similar to node 2 than it is to node 1. (c) With attention masking, the attention is limited to direct neighbors, so now $(3, 1)$ has the highest attention.

- **Challenge 2: Heavy Long-Tailed Distribution.** Most real-world graphs follow a long-tailed distribution in the node degrees where a majority of nodes have low degrees, and also follow a long-tailed class distribution meaning that the number of nodes in certain classes significantly dominates the other classes (Park et al., 2021; Zhao et al., 2021). It is critical to eliminate the bias towards the dominant classes that are introduced by these distribution properties of graphs since these biases can result in poor performance on nodes from underrepresented classes. However, the attention mechanism in existing GTs makes the model focus on certain parts of the input and could neglect the features relevant to underrepresented classes.

  **Observation: Combining a direct neighbor-attended GT with an MP-GNN can generally correct bias.** The attention layer in GTs tends to generate high attention weights for a few dominant nodes that are similar to the target nodes while ignoring underrepresented nodes in the neighborhood that are less similar. In contrast to this, the layers in MP-GNN can aggregate information from direct neighbors regardless of their similarity. Thus, their combination leads to a

more comprehensive view of the neighborhood information collection, as shown in Figure 2.

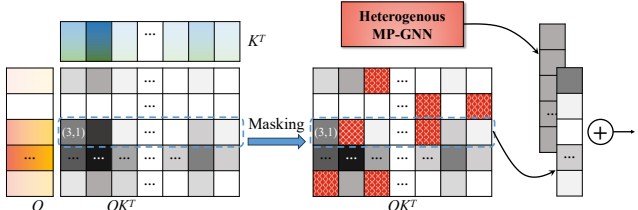

Figure 2: The attention mechanism may overlook the importance of certain nodes in the neighborhood due to their similarity. A similar toy example as Figure 1 is shown. Let's focus on row 3. After direct neighbor masking, attention at $(3, 1)$ dominates attention in all the other cells (other neighbors). Messages are not effectively aggregated from the other neighbors. This may be harmful when $(3, 1)$ belongs to a dominant class or a node with an inconsistent label. Instead, neighborhood information can be additionally aggregated from a heterogeneous MP-GNN module and combined to form a more comprehensive view of the neighborhood.

- **Challenge 3: Graph Heterophily.** Many of the existing works on GNN models share a vital assumption of graph homophily, i.e., locally connected nodes should be similar to each other in terms of their features and labels. However, recent research has shown the limitations of graph models in dealing with graphs with heterophily, where nodes with similar features or common connections have different classes (Zhu et al., 2021; Zheng et al., 2022). This severely affects the GT attention mechanism's performance, since it focuses on the part of input that has higher similarity. Adding in an MP-GNN may alleviate this issue by allowing communication between neighbors from different classes.

## 3 UNIFIED GRAPH TRANSFORMER FRAMEWORK (UNIFIEDGT)

In this section, to better analyze and improve the existing GTs for real-world practices, we first systematically summarize the existing GTs by introducing a general unified graph transformer framework, UnifiedGT (see Figure 3). Unified-GT consists of five modular ingredients: (i) graph sampling, (ii) structural prior, (iii) attention calculation, (iv) MP-GNN injection, and (v) fully connected layer. Most of the existing GTs can be easily included in this framework with various designs of each modular ingredient, as shown in Table 1.

Table 1: This table shows different components of GT methods. We list a few representative existing GT methods in the upper part of the table and indicate whether they are for designed for homogeneous or heterogeneous graphs. The lower part of the table shows our proposed new variants (marked with *).

| Methods | Sampling | Attention | Structural Prior | MP-GNN Composition | FFN |
|---|---|---|---|---|---|
| HGT (Hetero.) (Hu et al., 2020) | HGSampling | EAttn | Direct Neighbor Masking | – | – |
| GraphGPS (Homo.) (Rampášek et al., 2022) | – | Plain | Positional Encodings | Parallel | Single |
| GraphTrans (Homo.) (Wu et al., 2021) | – | Plain | – | Prefixed | Single |
| GraphiT (Homo.) (Mialon et al., 2021) | – | Plain | Kernel-based Masking | Prefixed | Single |
| ParDNTrans (Hetero.) (*) | HGSampling | NAttn | Direct Neighbor | Parallel | Type |

- **Graph sampling**. Graph sampling is an important step in dealing with large graphs. However, since existing GTs mainly focus on small graphs (fewer than thousands of nodes), graph sampling methods are rarely adopted. To extend GTs to large graphs, we provide three standard sampling options: (1) *Neighbor Sampling* (Hamilton et al., 2017), (2) *GraphSAINT sampling* (Zeng et al., 2019) for homogenous graphs, and (3) *HGSampling* (Hu et al., 2020) for heterogeneous graphs. HGSampling is a layer-based sampling method that can maintain a similar number of sampled nodes and edges for each type.
- **Structural prior**. Capturing the structure information is important for graph representation learning. Existing GTs use positional encodings (PE) to store structural information on each node at a local, relative, or global level, and they focus on homogeneous graphs. PEs in existing GTs either focus on local or relative structural information when processing large graphs. Examples of local PE include annotating node features with node degree (Ying et al., 2021) or the landing probability of an $m$-step *random walk* (Dwivedi et al., 2021). Examples of relative structural encodings include pairwise *shortest-path distances* (Li et al., 2020; Ying et al., 2021) or Boolean values to indicate whether two nodes are in the same structure (Bodnar et al., 2021). However, as we

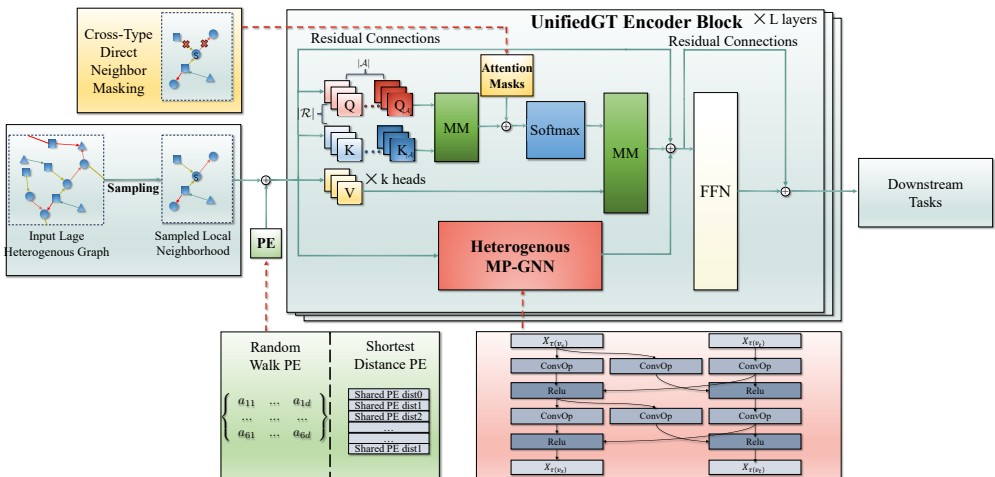

Figure 3: Overall framework, consisting of (i) graph sampling, (ii) structural prior injection (through injected positional encodings and attention mask), (iii) attention calculation, (iv) incorporation of local message-passing with long-range attention (heterogeneous MP-GNN modules), and (v) fully-connected layer (feed-forward network, FFN). We denote MM as matrix multiplication.

discussed in Challenge 1 in Section 2.1, PEs in existing GTs cannot capture heterogeneous information. To extend existing GTs to heterogeneous graphs, we propose a novel design for learning graph structure, *Direct Neighbor Attention Masking*, which will be described in Section 4.

- **Attention Calculation**. Existing GTs have predominantly been applied on homogeneous graphs, where the attention is calculated by pairwise dot product on the projected key and query vectors, assuming both vectors fall in the same projection space (Vaswani et al., 2017b). We call this *Plain Attention Calculation*. To generalize GTs to heterogeneous graphs, we propose two novel attention calculation methods, one based on edge-type (*EAttn*) and another is based on node-type (*NAttn*). We will elaborate on these methods in Section 4 in detail.

- **MP-GNN injection**. As discussed in Challenge 2 in Section 2.1, transformers can neglect underrepresented, but important nodes. Injecting an MP-GNN into a GT architecture can lead to a more comprehensive view of the neighborhood information collection. Generally, according to the relative position between MP-GNN layers and transformer layers, there are several injection schemes: prefixed connection, parallel connection, and more, which will be detailed in Section 4.

- **FFN**. The feed-forward network (FFN) is a typical component of transformer architectures Vaswani et al. (2017b), which plays an important role as a semantic mixer in the so-called Metaformer architecture Yu et al. (2022). The FFN introduces additional non-linear transformations, which can help the model capture more complex patterns and relationships in the data, allowing for more discriminative representations. Though some GTs neglect FFN, We added it as an important component in UnifiedGTs with two designs: *single FFN* and *type-specific FFNs*. A single FFN is a shared MLP network that is uniformly applied on all node embeddings from all of the available node types. It assumes the node embeddings are located in a unified semantic space, so they can share a unified FFN. A type-specific FFN assumes nodes of different types are in different semantic spaces, and allows modeling of richer relationships in the semantic space of each individual node type. Our empirical experiments show the effectiveness of FFN.

## 4 DESIGN CHOICES IN UNIFIEDGT

In this section, we introduce the details of the proposed novel ingredients (marked with * in Table 1), which are important parts of the UnifiedGT that enable GTs to deal with heterogeneous graphs.

### 4.1 ATTENTION CALCULATION

GTs with plain attention calculations have predominantly been applied on homogeneous graphs. To generalize GTs on heterogeneous graphs, we have introduced several enhancements over the plain attention calculation. In this framework, we propose two designs of attention calculation for heterogeneity considering two primary elements in graph, edge and node.

The first design, **EAttn**, an edge-type based attention, is inspired by Hu et al. (2020), which calculates the pairwise attention based on edge type (meta relation). Different sets of projections matrices $\boldsymbol{W}_{K_{\phi(e)}}$, $\boldsymbol{W}_{Q_{\phi(e)}}$, and $\boldsymbol{W}_{V_{\phi(e)}}$ are applied for each edge type (Equation (1)). For example, two edges type $\langle \text{author}, \text{write}, \text{paper} \rangle$ and $\langle \text{paper}, \text{cites}, \text{paper} \rangle$ have two different sets of projections matrices, which project the source node encodings $\boldsymbol{X}_{\tau(s)}^{(l)}$ and target node encodings $\boldsymbol{X}_{\tau(t)}^{(l)}$ into different semantic spaces. Attention weights are then calculated based on the pairwise dot products between the projected source-node type and target node-type encodings (Equation (2)).

$$\boldsymbol{Q}_{\phi(e)} = \boldsymbol{X}_{\tau(t)}^{(l)} \boldsymbol{W}_{Q_{\phi(e)}}, \ \boldsymbol{K}_{\phi(e)} = \boldsymbol{X}_{\tau(s)}^{(l)} \boldsymbol{W}_{K_{\phi(e)}}, \ \boldsymbol{V}_{\phi(e)} = \boldsymbol{X}_{\tau(s)}^{(l)} \boldsymbol{W}_{V_{\phi(e)}} \tag{1}$$

$$\text{Attn}_{\phi(e)} \left( \boldsymbol{X}_{\tau(t)}^{(l)} \right) = \text{softmax} \left( \frac{\boldsymbol{Q}_{\tau(t)} \boldsymbol{K}_{\tau(s)}^{T}}{\sqrt{d_k}} \right) \boldsymbol{V} \tag{2}$$

The second design, **NAttn**, stacks all of the different types of nodes together, i.e., $\boldsymbol{X}^{(l)} = [\boldsymbol{X}_1^{(l)T}, \boldsymbol{X}_2^{(l)T}, \ldots, \boldsymbol{X}_k^{(l)T}]^T$, where $\boldsymbol{X}_k^{(l)}$ is the node encoding matrix of a certain node type in $\mathcal{A}$. Different sets of projections matrices $\boldsymbol{W}_{K_i}$, $\boldsymbol{W}_{Q_i}$, and $\boldsymbol{W}_{V_i}$ are applied to different node types (Equation (3)). Attention weight is calculated following the pairwise dot product in Equation (5), where each node attends to all of the other nodes in the graph regardless of their type. Typical heterogeneous graph neural networks only pass messages through specified meta-relations (Schlichtkrull et al., 2018; Hu et al., 2020), while our proposed *NAttn* enables *cross-node-type communication*, where a single node can communicate with all other nodes of different types simultaneously.

$$\boldsymbol{Q} = [(\boldsymbol{X}_1^{(l)} \boldsymbol{W}_{Q_1})^T, (\boldsymbol{X}_2^{(l)} \boldsymbol{W}_{Q_2})^T, \ldots, (\boldsymbol{X}_k^{(l)} \boldsymbol{W}_{Q_k})^T]^T$$
$$\boldsymbol{K} = [(\boldsymbol{X}_1^{(l)} \boldsymbol{W}_{K_1})^T, (\boldsymbol{X}_2^{(l)} \boldsymbol{W}_{K_2})^T, \ldots, (\boldsymbol{X}_k^{(l)} \boldsymbol{W}_{K_k})^T]^T \tag{3}$$
$$\boldsymbol{V} = [(\boldsymbol{X}_1^{(l)} \boldsymbol{W}_{V_1})^T, (\boldsymbol{X}_2^{(l)} \boldsymbol{W}_{V_2})^T, \ldots, (\boldsymbol{X}_k^{(l)} \boldsymbol{W}_{V_k})^T]^T$$

*EAttn* allows communication on specified meta-relations and is suitable for handling the multiple relation types that appear in heterogeneous graphs. *NAttn* enables richer communication between node types, and it's suitable for heterogeneous graphs with limited predefined relation types. Note that when the input graph is homogeneous, there is only a single type of node and edge, and *NAttn* and *EAttn* are both equivalent to a plain attention calculation—they both contain a single set of projection matrices $\boldsymbol{W}_Q$, $\boldsymbol{W}_K$, and $\boldsymbol{W}_V$.

## 4.2 Explicit Structural Prior Injection Through Attention Masking

Although using positional encodings can inject graph structural priors into GTs, they cannot capture heterogeneous information. Prior work has proposed directly injecting graph priors through an added attention bias mechanism (Dwivedi & Bresson, 2020; Mialon et al., 2021), which can be formulated as Equation (4). Since the attention scores are normalized by the softmax function, adding an infinitely small bias to the attention scores between a pair of nodes $i$ and $j$ (force $\boldsymbol{B}(i,j) = -\infty$) equivalently masks out the attention between them. Inspired by the prior work using attention bias and selective attention to the direct neighbor (Hu et al., 2020; Yao et al., 2020), we propose a *cross-type direct neighbor attention masking* mechanism that does not require a fundamental change of existing GTs but can restrict them to focus on the important direct neighbors. We name it *cross-type* because it allows a single node to attend to its direct neighbors regardless of their node type.

$$\boldsymbol{A} = \boldsymbol{Q}\boldsymbol{K}^T / \sqrt{d_k} + \boldsymbol{B} \tag{4}$$

## 4.3 Composition of Local Message Passing and Long-range Attention

Transformers are useful when they can pay special attention to the informative nodes but can neglect underrepresented, but important nodes when there are long-tailed distributions of classes and graph heterophily. Inspired by the success of applying an MP-GNN as an auxiliary module in GTs, we incorporate the MP-GNN into our unified GT framework to help aggregate information from underrepresented nodes. Generally, according to the relative position between GNN layers and transformer layers, there are several composition schemes: prefixed and parallel connections.

A prefixed MP-GNN connection, the most frequently adopted method, performs a few layers of message passing using an MP-GNN before the node embedding enters the transformer. It allows

the MP-GNN to perform message passing from multi-hop neighbors via multilayer message passing and therefore mixes in more neighbor information in the node embedding. For example, Graph-Trans (Wu et al., 2021) employs such a configuration. The GNN layer learns local representations of the structure of a node's immediate neighborhood, while the transformer computes all pairwise node interactions in a position-agnostic fashion.

A parallel connection performs message passing using an MP-GNN and applies a transformer in parallel, summing together the output. It lets the MP-GNN focus on local message passing, while the long-range attention mechanism in the transformer accounts for the long-range neighbors that have a similar node embedding. For example, GraphGPS Rampášek et al. (2022) uses a hybrid MP-GNN and transformer with a parallel connection and demonstrates competitive results.

Furthermore, by simply combining the above options, there can be more connection schemes. In our study, we also experiment with one new scheme, prefixed + parallel connection, that hasn't been applied to existing GTs. It demonstrates the modularity of our framework.

## 5 EXPERIMENTS

### 5.1 GENERAL SETUP

**Datasets.** We study the Paper-Venue classification task in the Open Academic Graph (OAG) dataset (Sinha et al., 2015; Tang et al., 2008; Zhang et al., 2019). This dataset includes papers from multiple fields, such as Computer Science (CS) and Engineering (Eng), spanning the years 1900 to 2019. We use the subset of data corresponding to the CS and Eng fields (see Table 2 for statistics of the dataset). We split the papers published up to 2016 as the training set, papers published in 2017 as the validation set, and papers published in 2018 and 2019 as the test set.

**Task and Evaluation.** The Paper-Venue task we used to evaluate the model is to predict which venue the paper is published at from around three thousand possible choices. We use different GNN methods to obtain the contextual node representation of the paper and use a softmax output layer to obtain its classification label. We report both the accuracy and normalized discounted cumulative gain (NDCG) scores, which are widely adopted performance metrics (Liu et al., 2009).

### 5.2 EXPERIMENTAL RESULTS

We firstly study the effectiveness of components in UnifiedGT.

**Attention calculation.** We evaluate the performance of the two proposed attention calculation methods and validate their effectiveness on heterogeneous graph learning. As shown in Table 4, the two variants, EAttn and NAttn with direct neighbor masking consistently outperform the baselines (including the state-of-the-art HGT model(Hu et al., 2020)), with a performance gap of between **2–4%** and **5–6%** in terms of accuracy and between **1.5–3%** and **3–4%** in terms of NDCG,

Table 2: The statistics of the OAG datasets in our experiments.

| Name | # Graph | # Nodes | # Edges | # Classes |
|---|---|---|---|---|
| OAG-CS | 1 | 1,112,691 | 13,776,284 | 3,515 |
| OAG-Eng | 1 | 929,315 | 6,175,392 | 3,958 |

Table 3: Comparison of various GNN methods and the proposed model on OAG dataset. DN stands for direct neighbor attention masking. DD stands for double depth. We highlight the top **first**, **second**, and **third** results.

| Methods | OAG-CS | | OAG-Eng | |
|---|---|---|---|---|
| | Accuracy(%) | NDCG(%) | Accuracy(%) | NDCG(%) |
| MLP | $7.56 \pm 0.21$ | $31.33 \pm 0.10$ | $19.96 \pm 0.22$ | $46.33 \pm 0.43$ |
| GCN | $24.17 \pm 0.54$ | $50.52 \pm 0.22$ | $39.15 \pm 0.33$ | $63.65 \pm 0.27$ |
| SAGE | $24.54 \pm 0.19$ | $50.65 \pm 0.09$ | $37.22 \pm 0.22$ | $62.61 \pm 0.12$ |
| SAGE (DD) | $23.69 \pm 0.39$ | $50.32 \pm 0.20$ | $37.42 \pm 0.08$ | $62.55 \pm 0.01$ |
| GAT | $22.54 \pm 0.10$ | $48.87 \pm 0.10$ | $36.20 \pm 0.44$ | $61.13 \pm 0.26$ |
| HGT | $23.65 \pm 0.42$ | $49.85 \pm 0.33$ | $37.07 \pm 0.18$ | $61.98 \pm 0.29$ |
| ParDNTrans | $28.98 \pm 0.27$ | $53.97 \pm 0.18$ | $41.53 \pm 0.12$ | $65.58 \pm 0.22$ |

respectively. Interestingly, we noticed that NAttn consistently outperforms EAttn. This may be ascribed to the ability of NAttn to communicate in a cross-type manner which is beneficial for this dataset while EAttn only allows communication in the limited pre-defined meta-relations. This demonstrates the effectiveness of the two proposed modules.

**Structural prior.** We evaluate the effect of two kinds of structural prior injection methods, positional encodings, and attention masking, and validate the effectiveness of attention masking in heterogeneous graph learning. From Table 5, it can be observed that positional encodings have different

Table 4: Comparison of the effect of EAttn and NAttn, attention masking, and the performance with different FFN modules. The shade of green color represents the mean value of metric in each cell. The highest metric of each group is highlighted in bold. **DN**, direct neighbor attention masking.

| | OAG-CS | | OAG-Eng | |
|---|---|---|---|---|
| **Methods** | **Accuracy(%)** | **NDCG(%)** | **Accuracy(%)** | **NDCG(%)** |
| EAttn-Full (Single FFN) | $11.60 \pm 2.99$ | $37.22 \pm 2.27$ | $21.72 \pm 0.71$ | $46.79 \pm 0.29$ |
| EAttn-Full (Type-specific FFN) | $10.01 \pm 2.15$ | $37.20 \pm 1.75$ | $21.86 \pm 0.52$ | $46.72 \pm 0.59$ |
| EAttn-DN (no FFN) | $26.45 \pm 0.49$ | $52.19 \pm 0.35$ | $39.88 \pm 0.42$ | $64.18 \pm 0.26$ |
| EAttn-DN (Single FFN) | $26.90 \pm 0.22$ | $52.70 \pm 0.17$ | $\mathbf{40.21 \pm 0.80}$ | $\mathbf{64.44 \pm 0.39}$ |
| EAttn-DN (Type-specific FFN) | $\mathbf{27.74 \pm 0.42}$ | $\mathbf{53.08 \pm 0.26}$ | $39.94 \pm 0.24$ | $64.38 \pm 0.20$ |
| NAttn-Full (Single FFN) | $21.00 \pm 2.33$ | $45.52 \pm 1.99$ | $26.99 \pm 0.26$ | $52.04 \pm 0.12$ |
| NAttn-Full (Type-specific FFN) | $22.97 \pm 2.50$ | $46.90 \pm 2.07$ | $27.02 \pm 0.46$ | $52.07 \pm 0.27$ |
| NAttn-DN (no FFN) | $29.15 \pm 0.31$ | $54.26 \pm 0.28$ | $41.16 \pm 0.17$ | $65.20 \pm 0.13$ |
| NAttn-DN (Single FFN) | $29.31 \pm 0.70$ | $54.03 \pm 0.35$ | $\mathbf{41.49 \pm 0.65}$ | $\mathbf{65.47 \pm 0.39}$ |
| NAttn-DN (Type-specific FFN) | $\mathbf{29.48 \pm 0.70}$ | $\mathbf{54.30 \pm 0.67}$ | $41.32 \pm 0.46$ | $65.40 \pm 0.24$ |

Table 5: Comparison of effectiveness utilizing positional encodings. All GTs use a type-specific FFN. **RWPE**, random-walk-based positional encoding; **SDPE**, shortest-distance-based positional encoding.

| | OAG-CS | | OAG-Eng | |
|---|---|---|---|---|
| **Methods** | **Accuracy(%)** | **NDCG(%)** | **Accuracy(%)** | **NDCG(%)** |
| EAttn-Full | $10.01 \pm 2.15$ | $37.20 \pm 1.75$ | $21.86 \pm 0.52$ | $46.72 \pm 0.59$ |
| EAttn-Full + RWPE | $7.25 \pm 0.29$ | $33.60 \pm 1.59$ | $20.72 \pm 0.65$ | $45.92 \pm 0.19$ |
| EAttn-Full + SDPE | $7.42 \pm 0.61$ | $35.35 \pm 0.73$ | $22.36 \pm 0.58$ | $48.0 \pm 0.39$ |
| NAttn-Full | $22.97 \pm 2.50$ | $46.90 \pm 2.07$ | $27.02 \pm 0.46$ | $52.07 \pm 0.27$ |
| NAttn-Full + RWPE | $24.82 \pm 1.33$ | $49.31 \pm 0.95$ | $27.14 \pm 0.20$ | $52.26 \pm 0.06$ |
| NAttn-Full + SDPE | $14.38 \pm 0.27$ | $42.36 \pm 0.89$ | $27.66 \pm 0.22$ | $52.86 \pm 0.20$ |

Table 6: Comparison of various GTs with different MP-GNN composition schemes. To ensure a fair comparison, we use SAGEConv as the MP-GNN. All GTs use type-specific FFN. We highlight the top first, second, and third results. The shade of green color represents the mean value of metric in each cell for easy comparison. **DN**, direct neighbor attention masking; **Full**, no attention masking.

| | OAG-CS | | OAG-Eng | |
|---|---|---|---|---|
| **Methods** | **Accuracy(%)** | **NDCG(%)** | **Accuracy(%)** | **NDCG(%)** |
| EAttn-Full + prefixed GNN | $13.06 \pm 8.87$ | $37.20 \pm 8.79$ | $7.0 \pm 5.56$ | $32.66 \pm 4.49$ |
| EAttn-Full + parallel GNN | $28.03 \pm 0.33$ | $53.11 \pm 0.19$ | $39.02 \pm 0.22$ | $63.47 \pm 0.07$ |
| EAttn-Full + prefixed/parallel GNN | $28.52 \pm 0.85$ | $53.45 \pm 0.36$ | $38.89 \pm 0.37$ | $63.55 \pm 0.12$ |
| NAttn-Full + prefixed GNN | $27.51 \pm 0.69$ | $52.74 \pm 0.42$ | $38.96 \pm 0.26$ | $63.4 \pm 0.11$ |
| NAttn-Full + parallel GNN | $30.50 \pm 0.99$ | $55.22 \pm 0.70$ | $40.74 \pm 0.04$ | $64.82 \pm 0.04$ |
| NAttn-Full + prefixed/parallel GNN | $27.99 \pm 0.67$ | $53.13 \pm 0.20$ | $39.48 \pm 0.46$ | $63.98 \pm 0.27$ |
| EAttn-DN + prefixed GNN | $26.22 \pm 0.44$ | $51.97 \pm 0.33$ | $40.07 \pm 0.25$ | $64.54 \pm 0.11$ |
| EAttn-DN + parallel GNN | $27.64 \pm 0.41$ | $53.11 \pm 0.31$ | $40.36 \pm 0.43$ | $64.65 \pm 0.19$ |
| EAttn-DN + prefixed/parallel GNN | $26.92 \pm 0.44$ | $52.54 \pm 0.17$ | $39.76 \pm 0.30$ | $64.48 \pm 0.03$ |
| NAttn-DN + prefixed GNN | $28.55 \pm 0.16$ | $53.85 \pm 0.16$ | $41.36 \pm 0.13$ | $65.47 \pm 0.04$ |
| NAttn-DN + parallel GNN | $28.98 \pm 0.27$ | $53.97 \pm 0.18$ | $41.53 \pm 0.12$ | $65.58 \pm 0.22$ |
| NAttn-DN + prefixed/parallel GNN | $28.22 \pm 0.08$ | $53.15 \pm 0.07$ | $40.74 \pm 0.32$ | $65.06 \pm 0.15$ |

effects – they degraded the performance when combined with EAttn, but marginally improved the performance in NAttn by up to 0.64%. In contrast, Table 4 shows that after applying direct neighbor attention masking, the performance of EAttn and NAttn are greatly improved by around **17%** and **7%** in terms of accuracy and **16%** and **8%** in terms of NDCG. The results illustrate that attention masking is an effective structural prior injection method for heterogeneous graph learning.

**MP-GNN Composition.** Next, we assess the contribution of MP-GNN composition and validate the combined architecture can lead to competitive results. We perform a study on three configurations for composing the auxiliary MP-GNN with a GT. From Table 6, we observe that the model with a parallel MP-GNN connection generally outperforms the other options. More interestingly, when a GT without attention masking combined with MP-GNN composition, its performance greatly increased (for example, compared with the 10.01% accuracy EAttn-Full and 22.97% accuracy NAttn-Full achieved in Table 4). This implies the MP-GNN composition also implicitly includes graph structure information that improves the performance of GT. To rule out the possibility that the performance improvements are simply the effect of a deeper GNN network, we also tested the performance of a SAGEConv-based GNN with double depth (see Table 3) so that both models have the same number of layers. We observed that simply increasing the model depth only leads to either comparable performance or worse performance. This demonstrates the MP-GNN composition did collaborate with the transformer modules and improve the performance of both.

**FFN.** Lastly, we examine the impact of FFN and our proposed variant (type-specific FFN) and validate their important role in graph transformers. From Table 4, compared with the methods with no FFN, the two options of FFN clearly boost the performance when combined with either EAttn or NAttn. Specifically, when using a type-specific FFN in the OAG-CS dataset, it increases the accuracy by **1.29%** when using EAttn. The choice of FFN is not unique for different datasets – the performance with a single FFN outperforms the one with type-specific FFN in the OAG-Eng dataset.

Overall, we summarize the experimental results of the proposed variant, *ParDNTrans*, and baselines in Table 3. The composition of the proposed variant can be found in Table 1. The results show that in terms of both metrics, the proposed variants can outperform all baselines. Particularly, our best model can outperform the state-of-the-art HGT (Hu et al., 2020) model on OAG-CS dataset by **5.33%**, **4.12%** and on OAG-Eng dataset by **4.46%**, **3.6%** in terms of accuracy and NDCG, respectively. They exhibit an improvement over the best baseline on the OAG-CS dataset of **4.44%**, **3.32%** and on the OAG-Eng dataset of **2.38%**, **1.93%** in terms of accuracy and NDCG, respectively. Furthermore, the variant we propose, which includes direct neighbor masking, demonstrates competitive performance on two heterogeneous graph datasets. The GT with direct neighbor attention masking manifests characteristics of sparse matrices. This offers the potential to harness the sparse matrix operations inherent in accelerator hardware, thereby facilitating training and inference processes. These findings suggest that computing the entirety of the dense attention matrix may not be necessary to attain optimal performance in graph learning.

# 6 RELATED WORKS

## 6.1 GRAPH TRANSFORMERS

Graph transformers Ying et al.; Min et al. (2022); Rampášek et al. (2022); Ying et al. (2021); Zhang et al. (2022); Rampášek et al. (2022); Hu et al. (2020); Yao et al. (2020); Kong et al. (2023) extend the transformative capabilities of conventional transformer architectures, which have made significant strides in both natural language processing Vaswani et al. (2017a); Kalyan et al. (2021); Zhang et al. (2022) and computer vision Han et al. (2023); Ranftl et al. (2021); Zhai et al. (2022). By using powerful attention mechanisms, they capture long-range dependencies in graph structures efficiently. Transformers overcome limitations in traditional MP-GNN Kipf & Welling (2016); Hamilton et al. (2017), such as over-smoothing and over-squashing Song et al. (2023). To deal with the idiosyncrasies of graph data, like local structure and relative positional information, specialized techniques such as graph-specific positional and structure encodings have been proposed Chen et al. (2022; 2023). Current research primarily focuses on homogeneous graphs Ying et al.; Kong et al. (2023) or specialized applications like heterogeneous molecule graphs Hu et al. (2020); Wang et al. (2019); Rampášek et al. (2022). Our work aims to bridge this gap by providing a graph transformer framework that can handle large heterogeneous graphs.

## 6.2 LEARNING ON LARGE-SCALE GRAPHS

The computational intricacies of learning on large-scale graphs frequently cause traditional MPNNs to run out of memory Gupta et al. (2021). To mitigate this, various solutions have been proposed. Graph partitioning Fjällström (1998) splits the graph into smaller, manageable sub-graphs to facilitate parallel computing Shao et al. (2022). Sampling methods Hu et al. (2020), such as node or edge sampling, reduce the computational load by focusing on representative subgraphs Hu & Lau (2013); Hamilton et al. (2017); Zeng et al. (2019). Mini-batching techniques strive for a balanced compromise between computational efficiency and learning efficacy Chiang et al. (2019). All of these strategies aim to optimize learning outcomes while preserving the graph's structural and feature distributions. This paper applies these strategies to graph transformers to perform learning on large heterogeneous graphs.

# 7 CONCLUSION

In this paper, we proposed the UnifiedGT framework for learning on large heterogeneous graphs, which includes various options for graph sampling, structural prior injection, attention calculation, and composition of local message-passing and long-range attention. Using UnifiedGT, we performed a systematic empirical study to investigate the critical components of GTs and broaden the understanding of the role of each component in overcoming the challenges in learning on large heterogeneous graphs. We conducted experiments on large citation networks in the Open Academic Graph dataset and showed that the best-performing method in UnifiedGT can outperform all of the current baselines. We hope to accelerate the research of graph transformers on large-scale heterogeneous (and homogeneous) graphs and move the field closer to a unified transformer architecture for graphs.

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

## A  APPENDIX

### A.1  PRELIMINARIES

**Heterogeneous Graphs.** A heterogeneous graph is defined as a directed graph $\mathcal{G} = (\mathcal{V}, \mathcal{E}, \mathcal{A}, \mathcal{R})$, where each node $v \in \mathcal{V}$ and each edge $e \in \mathcal{E}$ is associated with their type mapping function $\tau(v) : V \to \mathcal{A}$ and $\phi(e) : E \to \mathcal{R}$, respectively.

**Meta Relation.** For an edge $e = (s, t)$ linked from source node $s$ to target node $t$, its meta relation is denoted as $\langle \tau(s), \phi(e), \tau(t) \rangle$. The classical meta path is defined as a sequence of such meta relations. To better model real-world heterogeneous graphs, we assume that there may exist multiple types of relations between different types of nodes. For example, in citation networks, there are different types of relations between the *author* and *paper* nodes, such as $\langle$author, is first author of, paper$\rangle$, $\langle$author, is last author of, paper$\rangle$, etc.

**Graph Transformers (GTs).** A transformer is a stack of alternating blocks of multi-head attention and fully connected feed-forward networks. Let $G$ be a graph with node feature matrix $\boldsymbol{X} \in \mathbb{R}^{n \times d}$. In each layer $l$ ($l > 0$), given the node feature matrix $\boldsymbol{X}^{(l)} \in \mathbb{R}^{n \times d}$, a single attention head computes

$$\text{Attn}\left(\boldsymbol{X}^{(l)}\right) = \text{softmax}\left(\frac{\boldsymbol{Q}\boldsymbol{K}^T}{\sqrt{d_k}}\right)\boldsymbol{V}, \tag{5}$$

where the softmax is applied row-wise, $d_k$ denotes the feature dimension of the matrices $\boldsymbol{Q}$ and $\boldsymbol{K}$, and $\boldsymbol{X}^{(0)} = \boldsymbol{X}$, the input node feature matrix. The matrices $\boldsymbol{Q}$, $\boldsymbol{K}$, and $\boldsymbol{V}$ are the result of projecting $\boldsymbol{X}^{(l)}$ linearly, i.e.,

$$\boldsymbol{Q} = \boldsymbol{X}^{(l)}\boldsymbol{W}_Q, \ \boldsymbol{K} = \boldsymbol{X}^{(l)}\boldsymbol{W}_K, \ \boldsymbol{V} = \boldsymbol{X}^{(l)}\boldsymbol{W}_V, \tag{6}$$

where $\boldsymbol{W}_Q, \boldsymbol{W}_K \in \mathbb{R}^{d \times d_K}$, and $\boldsymbol{W}_V \in \mathbb{R}^{d \times d}$. Multi-head attention $\text{MultiHead}(\boldsymbol{X}^{(l)})$ concatenates several attention heads. By combining the above with additional residual connections and normalization, the transformer layer updates features $\boldsymbol{X}^{(l)}$ through

$$\hat{\boldsymbol{X}}^{(l+1)} = \text{MultiHead}\left(\boldsymbol{X}^{(l)}\right) + \boldsymbol{X}^{(l)} \tag{7}$$

$$\boldsymbol{X}^{(l+1)} = \text{FFN}\left(\hat{\boldsymbol{X}}^{(l+1)}\right) + \boldsymbol{X}^{(l)} = \left[\sigma\left(\hat{\boldsymbol{X}}^{(l+1)}\boldsymbol{W}_1 + \boldsymbol{B}_1\right)\boldsymbol{W}_2 + \boldsymbol{B}_2\right] + \boldsymbol{X}^{(l)}, \tag{8}$$

where $\sigma$ refers to the activation function, and $\boldsymbol{W}_1 \in \mathbb{R}^{d \times d_f}, \boldsymbol{B}_1 \in \mathbb{R}^{d_f}, \boldsymbol{W}_2 \in \mathbb{R}^{d_f \times d}, \boldsymbol{B}_2 \in \mathbb{R}^d$ are trainable parameters in the feedforward network (FFN) layer.

## A.2 LONG-TAILED DISTRIBUTION IN LARGE GRAPHS

Figure 4 demonstrates the long-tailed distribution on the OAG-CS dataset.

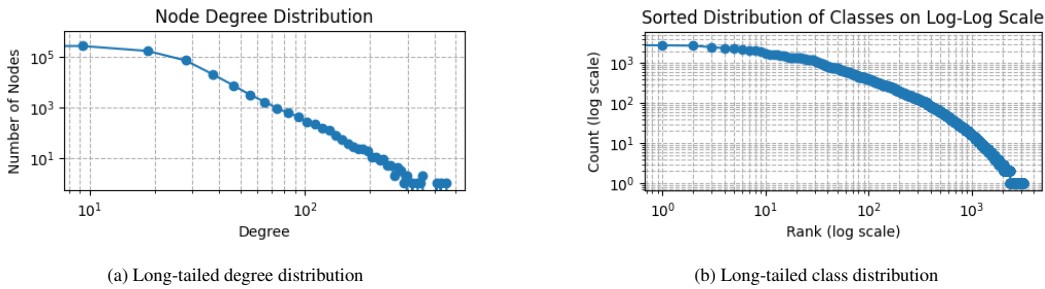

(a) Long-tailed degree distribution    (b) Long-tailed class distribution

Figure 4: Long-tailed degree and class distributions in the heterogeneous OAG-CS graph.

## A.3 HARDWARE

All the neural network training and data processing were performed on the Satori IBM Power9 cluster at the Massachusetts Institute of Technology. It is a GPU dense, high-performance Power 9 system developed as a collaboration between MIT and IBM. It has 64 1TB memory Power 9 nodes, each host 128 CPU cores that can run up to 3.8GHz and four Nvidia V100 GPU cards with 32GB dedicated memory. Within a node, GPUs are linked by an NVLink2 network that supports nearly 200GB/s bi-directional transfer between GPUs which enables powerful multi-card computation and machine learning. All processing was done by using Pytorch 1.12.1 in Python 3.9.

