# OpenReview forum: "UnifiedGT: Exploring the Effective Ingredients of Transformers in Large Graphs"
_ICLR.cc/2024/Conference — ICLR 2024 Conference Withdrawn Submission_

### Official Review · Reviewer_h6Lj · 2023-10-28

**Soundness:** 2 fair
**Presentation:** 2 fair
**Contribution:** 2 fair
**Rating:** 3
**Confidence:** 5

**Summary:**

This paper proposes a modular unified GT framework for graph representation learning, which consists of five major components. The proposed framework is claimed to be effective at handling both large-scale heterogeneous and homogeneous graph data, where the best-performing method, ParDNTrans, boosts accuracy by 4.5-5.3% over the state-of-the-art graph transformer.

**Strengths:**

1. The authors did a thorough study on various components of graph transformers, especially for heterogeneous graphs.

2. Extending graph transformers to large and heterogeneous graph is an important problem, and the issue addressed in this paper is interesting.

3. The authos clearly pointed out issues (Information Heterogeneity, Heavy Long-Tailed Distribution, Graph Heterophily) in exising graph-structured learning in the introduction

**Weaknesses:**

1. First of all, this paper is weak in experiments for the following two reasons. First, compasions are only performed on a single dataset, OAG, and it is unclear how their findings generalize to other large and heterogeneous graphs. Second, baselines are not properly selected. For example,  in Table 3, the second, third best performance models are GCN or SAGE (graphSAGE?), while these two models are not specifically designed for heterogeneous graphs. Some latest heterogeneous graph models to be added.

2. Though this paper had a thorough study to various components of GTs, disappointedly, key designs are from exisiting results, e.g., combining GT and gnn (graphGPS, graphTrans), heterogeneous graph (HGT),  attention bias mechanism (graphormer).

3. It is claimed that the model can deal with graph heterophily, but without supporting evidence.

**Questions:**

1. How the model performs on other large and heterogeneous graphs, and compared with some latest GTs?

2. Any metric measuring the heterophily performance?

3. What is the # of parameters of the proposed model and how it is comparable to baselines?

4. This title "LARGE GRAPHS" is a bit misleading. After reading the paper, I think it is more about extending GTs to heterogeneous graphs. Also, UNIFIEDGT is kind of overclaim, it is not discussed how existing GTs fall in the proposed framework.

5. I suggest put Table 3 after Table 6, consistent with the text

---

### Official Review · Reviewer_ybig · 2023-10-31

**Soundness:** 3 good
**Presentation:** 2 fair
**Contribution:** 2 fair
**Rating:** 3
**Confidence:** 3

**Summary:**

The paper articulates a design space for graph transformers (GTs) including (i) sampling, (ii) graph prior, (ii) attention computation, (iv) local message-passing and long-range information access, and (v) FF layers.  The authors suggest suitable choices for these can yield task-specific metric improvements over current GTs by several percent.

**Strengths:**

S1. The paper proposes a clear organization of the GT design space.

S2. The paper's use of figures to articulate challenges helps presentation significantly.

S3. The reported results from the method (ParDNTrans) seem to outperform existing GT methods.

**Weaknesses:**

W1. The paper claims several challenges which their proposals would tackle, without some empirical demonstrations of their existence/validity.

W2. The paper seems better fit for a survey/benchmarks paper as it does not seem to claim any methodological contribution but mostly articulates a design space.  This design space itself could be better explored over a longer paper which focuses on very careful ablations with each of the design choices proposed in Section 4.

W3. The paper seems to try to tackle a few issues at the same time (e.g. large graphs, heterogeneity, complex design space) which lead to diluted focus, and lack of clarity on what the core contribution of the work is.

**Questions:**

- Sec 2.1: "existing GTs inject graph priors ... through ... positional encodings, based on which attention scores ... elements located next to each other will have higher attention to each other" --> my understanding is that PEs allow distinguishing token position, but is there strong evidence that they induce higher similarity between adjacent positions?

- Challenge 2 is somewhat vague / ambiguous.  It covers two topics of heavy long-tailed degree distributions, as well as a class distribution (maybe relevant to tasks where nodes have associated classes, which is somewhat independent of model architecture).  Better separation of the discussion of these two topics and how they relate to GTs would be clearer.

- Is there supposed to be an observation from Challenge 3 (like there is from 1 and 2?)

- Evidences of the reality/presence of the challenges discussed in Section 2 would strongly help motivate this work.  Design elements being structured around philosophical challenges without demonstration of those challenges actually manifesting in practice makes it difficult to tie solution to problem.

- I appreciate the authors' diligence to organizing components of GTs which might help define a space of architectural choices.  However, this might be better fit for a benchmarks paper or a survey.  Several of the proposals in this manuscript, e.g attention calculation or MP-GNN injection or graph sampling, or structural prior are all choices which have been discussed or introduced individually in relevant works (as Table 1 refers).

- Sec 4.2 details a strategy for masking attention which involves adding a bias to a matrix B with N^2 computed elements based on all-pairs attention.  This would be slow for very large graphs.  Is there a reason we would opt for this instead of e.g. only implementing the attention logic over pairs of nodes which are connected?

- MPNNs are known to suffer scalability challenges for large-scale graphs owing to neighbor explosion issues as several past works have highlighted.  Several works [1, 2] undertake efforts to move away or minimize MPNN usage for large-scale applications where training or inference may suffer given this data dependency.  I am curious about the choice to inject MPNNs (Sec 4.3) as an ingredient towards scaling GTs to large graphs as a result.

- The graphs used for benchmarking (Table 2) are not particularly large in the context of available datasets (e.g. ogbn-products, ogbn-papers100m).

- Experimental results seem to focus on the discussed variants EAttn and NAttn, but it is not clear that is a key contribution of this work (if it is, it should be highlighted earlier in the intro) -- it sounds like these are mainly derived from Hu et al (2020).

- It is unclear what parts of the proposed model ParDNTrans are tied to methodological contributions this work aims to put forward -- it seems the components which comprise ParDNTrans are the components in the "UnifiedGT" design space which achieve the best performance, but it is not too clear why these components are critical parts of the design space.

- The paper misses several related large-scale graph learning works, including e.g. pretraining [4], embedding reuse [3], distillation [1, 5].  There is comparatively a lot of literature in large-scale graph learning in the MPNN space compared to GTs, admittedly.

[1] Graph-less Neural Networks: Teaching Old MLPs New Tricks via Distillation (ICLR'22)

[2] Decoupling the depth and scope of graph neural networks (NeurIPS'21)

[3] Gnnautoscale: Scalable and expressive graph neural networks via historical embeddings (ICML'21)

[4] MLPInit: Embarrassingly Simple GNN Training Acceleration with MLP Initialization (ICLR'23)

[5] Linkless Link Prediction via Relational Distillation (ICLR'23)

Typos:

- 2.1 "Challlenges" -> "Challenges"

---

### Official Review · Reviewer_Fnxk · 2023-11-01

**Soundness:** 3 good
**Presentation:** 3 good
**Contribution:** 2 fair
**Rating:** 5
**Confidence:** 4

**Summary:**

The paper proposes a unified modular graph transformer framework (UnifiedGT) to systematically study the key components of graph transformers on large heterogeneous graphs. UnifiedGT consists of 5 components: graph sampling, structural prior injection, attention calculation, message passing GNN injection, and fully connected layers. It introduces new methods like cross-type direct neighbor attention masking and edge/node-type based attention to handle graph heterogeneity. Experiments on the Open Academic Graph dataset demonstrate that explicit neighbor masking is more effective than positional encoding for injecting structure. Combining message passing GNNs with transformers boosts performance by aggregating underrepresented nodes. The proposed UnifiedGT framework with a parallel connected GNN and direct neighbor attention masking (ParDNTrans) improves accuracy over state-of-the-art methods.

**Strengths:**

- Proposes a modular framework to systematically study graph transformer components on large heterogeneous graphs
- Introduces effective new techniques like cross-type neighbor masking and type-based attention for heterogeneity
- Empirical study provides insights about critical ingredients like attention masking, MP-GNN injection, and FFN
- Achieves state-of-the-art performance on real-world citation network dataset

**Weaknesses:**

- The proposed techniques are evaluated on only one kind of dataset (academic citations)
- Unclear how the techniques would generalize to other heterogeneous graph tasks
- Lacks theoretical analysis to justify design choices

**Questions:**

- How do the techniques compare if evaluated on other heterogeneous graph datasets?
- Is there a theoretical justification for why direct neighbor masking is more effective than positional encodings?
- Could the techniques proposed generalize to homogeneous graphs as well?

---

### Official Review · Reviewer_psLQ · 2023-11-02

**Soundness:** 2 fair
**Presentation:** 2 fair
**Contribution:** 2 fair
**Rating:** 3
**Confidence:** 4

**Summary:**

This paper proposes a unified modular graph transformer framework, UnifiedGT, with configurable components for graph sampling, structural prior injection, attention calculation, incorporation of local message-passing with long-range attention, and fully connected layer. The authors also propose two new attention mechanisms for heterogeneous graph transformer and cross-type direct neighbor masking, which allows the GT to attend to direct neighbors of various node types. They empirically show that their best GTs, ParDNTrans, is able to boost accuracy by 4.5–5.3% over the state-of-the-art graph transformer.

**Strengths:**

1. This paper is overall clearly written and easy to follow.
2. The authors provided a good summarization of multiple existing GT methods and also unified them into a shared design space for better GT designs.
3. The challenges and observations provided in Sec. 2.1 are insightful and valuable for future works in designing GTs.

**Weaknesses:**

1. The phrase "large-scale" is mentioned everywhere in this paper from title to its body. However, it's missing in the experiments section, the authors only evaluated on two not very "large-scale" datasets. I'd suggest the authors to evaluate on more larger datasets, such as OGB-products or even OGB-papers100M.
2. Although the authors have successfully unified several graph transformer methods into the proposed unified framework, I found two very relevant works standing outside of the proposed unified framework: NAGphormer [1] and GOAT [2].
3. The design choices in UnifiedGT that are discussed in Sec. 4 seem were all from existing methods.

[1] NAGphormer: A Tokenized Graph Transformer for Node Classification in Large Graphs, ICLR 2023 \
[2] GOAT: A Global Transformer on Large-scale Graphs, ICML 2023

**Questions:**

please refer to the weaknesses above